# Application of the JA-CHRODIS Integrated Multimorbidity Care Model (IMCM) to a Case Study of Diabetes and Mental Health

**DOI:** 10.3390/ijerph16245151

**Published:** 2019-12-17

**Authors:** Maria João Forjaz, Carmen Rodriguez-Blazquez, Inmaculada Guerrero-Fernández de Alba, Antonio Gimeno-Miguel, Kevin Bliek-Bueno, Alexandra Prados-Torres

**Affiliations:** 1Department of Epidemiology and Biostatistics, National School of Public Health and REDISSEC. Carlos III Institute of Health, 28029 Madrid, Spain; jforjaz@isciii.es; 2National Centre of Epidemiology and CIBERNED. Carlos III Institute of Health, 28029 Madrid, Spain; crodb@isciii.es; 3EpiChron Research Group, IIS Aragón, Teaching Unit of Preventive Medicine and Public Health, 50009 Zaragoza, Spain; kevinbliek@gmail.com; 4EpiChron Research Group, Aragon Health Sciences Institute, IIS Aragón, REDISSEC, Miguel Servet University Hospital, 50009 Zaragoza, Spain; agimenomi.iacs@aragon.es (A.G.-M.); sprados.iacs@aragon.es (A.P.-T.)

**Keywords:** multimorbidity, chrodis, Integrated Multimorbidity Care Model, comorbidity, chronic diseases, health promotion, disease prevention, implementation research, public health

## Abstract

The Integrated Multimorbidity Care Model (IMCM), developed by the Joint Action on Chronic Diseases and Promoting Healthy Ageing across the Life Cycle (JA-CHRODIS), proposes a set of 16 multidimensional components (i.e., recommendations) to improve the care of persons with multimorbidity in Europe. This study aimed at analyzing the potential applicability of the IMCM. We followed a qualitative approach that comprised two phases: (1) The design of a case study based on empirical clinical data, which consisted of a hypothetical woman with multimorbidity, type 2 diabetes mellitus, mental health, and associated social problems, and (2) the creation of a consensus group to gather the opinions of a multidisciplinary group of experts and consider the potential applicability of the IMCM to our case study. Experts described how care should be delivered to this patient according to each model component, suggested the use of specific rating scales and tools to assess her needs in a comprehensive and regular way, and pointed our crucial health and social resources to improve her care process. Experts also highlighted patient-centered, integrated and tailored care as one of the keystones of quality healthcare. Our results suggest that the IMCM is applicable in complex patients with multimorbidity.

## 1. Introduction

The care of patients with chronic diseases has become one of the most important issues for health organizations, as it leads to an important healthcare burden with up to 59% of deaths being caused by chronic diseases worldwide [1]. Multimorbidity, defined as the presence of two or more chronic diseases coexisting in the same person, represents a major challenge for public health, as it is becoming more and more prevalent in most European countries and is associated to negative health outcomes and increased costs for health systems [2,3,4]. The most frequently associated adverse outcomes to multimorbidity include lower quality of life, higher treatment burden (i.e., polypharmacy), higher risk of mortality, adverse drug events, and inappropriate use of health services, including unplanned and emergency care [4,5]. Multimorbidity is the most prevalent chronic condition, especially in older adults, reaching up to 90% of people over 65 years of age [6,7,8].

The design of care models for people with multimorbidity is becoming a priority for most healthcare systems, as they are still mainly oriented towards acute rather than chronic disease care [9,10]. The models designed to meet the needs of these patients require a comprehensive approach and the reorientation of healthcare systems. At present, specific care pathways for multimorbidity are scarce, not standardized, and have limited evidence of effectiveness [10]. In order to face these complex deficiencies, a multidimensional transformation of medical attention towards a patient-focused system would be necessary [11,12].

The Joint Action on Chronic Diseases and Promoting Healthy Ageing across the Life Cycle (JA-CHRODIS) brought together over 70 partners from 24 EU Member States aiming at minimizing the burden of chronic diseases and the impact of multimorbidity using the best knowledge currently available. In the absence of a specific care model capable of addressing the complex challenge that multimorbid patients represent, JA-CHRODIS recently developed the Integrated Multimorbidity Care Model (IMCM). This model identified a set of common standardized components for the care of patients with multimorbidity to be applied in different European healthcare systems [13].

The development of the JA-CHRODIS IMCM involved the collaboration of experts from different countries who identified a total of 20 key components in the delivery of care to multimorbid patients based on the systematic review conducted by Hopman et al. 2015 [10]. Subsequently, the expert group analyzed the relevance of the components for the integrated care of these patients, and finally selected 16 key components and grouped them into five areas: delivery of care, decision support, self-management support, information systems and technology, and social and community resources [14]. However, this theoretical model has not yet been implemented in real life conditions. In this regard, a study of its applicability would be of interest to facilitate the implementation of the model in regular clinical practice.

The main objective of this study was to analyze the potential applicability of the IMCM in a hypothetical multimorbidity case study with highly prevalent conditions, such as diabetes and mental health issues, and to describe the elements that need to be considered to apply each of the components of the model and facilitate its actual implementation in daily clinical practice.

## 2. Materials and Methods

This study followed a qualitative methodology consisting of two consecutive phases. The first step was to design a case study of a realistic and hypothetical woman with multimorbidity (‘Maria’s case’). Then, we distributed the case study among a group of experts from different countries and collected, analyzed, and summarized their opinions on the potential applicability of the IMCM to this specific case.

### 2.1. Maria’s Case

We developed this case study based on empirical data from multimorbidity studies containing population-based information from real healthcare registries [4,5]. Information on socio-demographic (i.e., age, gender, marital status, education level, urban/rural setting, employment status, number of children, caregiving of grandchildren) and clinical characteristics (i.e., number and type of chronic health conditions, mobility, sleep, obesity, healthcare service utilization, quality of life, self-rated health and activity levels) of patients with type 2 diabetes mellitus and mental health issues was gathered. To do so, the CHRODIS core team, comprising a group of eight JA-CHRODIS members from Work Packages WP6 (Multimorbidity) and WP7 (Diabetes), consulted the Survey of Health, Ageing and Retirement in Europe (SHARE) Wave 5 dataset [15]. The case was about a fictional female patient with multimorbidity, named Maria, described using detailed information on her socio-demographic, clinical, social, psychological, and family characteristics, as well as her main barriers and her use of health resources (Appendix A).

### 2.2. Collection and Analysis of Expert Opinions

The CHRODIS core team developed a questionnaire (Appendix A) to be distributed by email among a group of experts from different countries. This questionnaire, which was written and answered in English, collected detailed information on how each of the IMCM components should be ideally applied to Maria’s case.

The members of the group of experts were selected using a convenience sampling method. The CHRODIS core team contacted by email potential respondent experts, suggested by members from WP6 and WP7, to answer the questionnaire. Eleven experts (of a total of 20 contacted) from eight countries (Croatia, 1; Italy, 1; Germany, 1; Lithuania, 1; Netherlands, 2; Slovenia, 1; Spain 1; United Kingdom, 3) agreed to participate and report on the relevance of the 16 IMCM components for the care of patients with multimorbidity. The group of experts included general practitioners (GP), physicians from different specialties (i.e., neurologists, geriatricians, internists, cardiologists, endocrinologists, and diabetes specialists), epidemiologists, psychologists, and representatives from the patient organization.

To decrease respondent burden, the CHRODIS core team decided to have each of the 16 components answered by experts from two different countries, following the scheme showed in Table 1. The experts were asked to express their preferences on the components to be assessed, and they were finally assigned to a specific component by the research team to assure a balanced distribution by country and that all the components were covered.

M.J.F., C.R.B., and A.P.T performed a qualitative content analysis of the questionnaires to determine the presence of the most frequent words, themes, or concepts regarding each section of the model, and summarized the answers given to each IMCM component, focusing especially on the common information provided by more than one expert.

We designed word cloud charts to offer a visual representation of the most frequently repeated words used by experts when answering the questionnaires. These charts display relevant words in varying sizes that scale-up proportionally with their frequency of appearance, and therefore offer an intuitive depiction of the most important concepts repeated by different experts. The processing of the questionnaires included a critical search for significant words from each of the five sections of the model. We first removed meaningless words that had no influence on the semantics of sentences and then eliminated stop words (e.g., that, same, she) as the final step of questionnaire pre-processing [16,17]. By combining the data from each questionnaire and merging our findings from all five sections of the model, we obtained the final representation of the global word cloud.

This investigation did not require the use of personal data from patients or participants and therefore the approval from an ethics committee was unnecessary.

## 3. Results

The word cloud chart combines our findings from all five sections of the model (Figure 1) and shows that the most relevant words, based on their frequency of appearance in the questionnaires, were “support” (37 times cited), “information” (21), “team” (22), “care” (20), “contact” (19), “primary” (19), “self-management” (14) and “multidisciplinary” (12). Evaluating experts cited the word “support” in every section of the model, while “team”, “family”, and “primary” appeared in three out of the five sections. The word cloud created for the “information systems and technology” section depicted information (17), support (10), system (7), monitoring (6), privacy (5), access (5), and confidentiality (5) as the most critical elements. The individual word clouds corresponding to the different sections of the model can be found in the Appendix A.

Below are the summarized answers given by the experts to each of the 16 components of the five IMCM sections regarding their potential applicability to Maria’s case.

### 3.1. Delivery of Care

#### 3.1.1. Component 1: Regular Comprehensive Assessment of Patients

Maria’s case necessarily requires an integrated intervention where several professionals collaboratively assess her medical and psychological conditions. Experts agreed that the geriatrician would play an important role, but other professionals such as GPs, psychologists, physiotherapists, endocrinologists, and neurologists were also identified. Maria should be assessed at the primary care center and, only if needed, by a specialist, at least every three months after the first assessment and every six months after stabilization. Regarding Maria’s individualized care plan, the geriatrician and the nurse were identified as the key responsible figures.

Experts agreed on the importance of interviewing the patient and her relatives to assess their preferences and available resources and pointed out electronic medical records and clinical interviews as crucial elements in the assessment of patient complexity. In addition, respondents identified some useful tools for the evaluation of geriatric conditions (e.g., International Resident Assessment Instrument; Comprehensive Geriatric Assessment tools), cognitive functions (e.g., Cognitive Behavioral Assessment 2.0; Mini Mental State Examination; Mental Deterioration Battery), cardiovascular risk, and risk of falls (e.g., Conley/Hendrich II/MORSE). Furthermore, for more specific conditions, experts suggested other resources such as screening for peripheral artery disease, University of Texas Diabetic Foot Screen, Short Physical Performance Battery, Geriatric Depression Scale, and the Epworth Somnolence Scale, among others [18,19,20,21,22,23,24,25,26,27]. In general, these tools could provide valuable information about medical, psychological, and functional conditions, as well as personal and social needs and resources. The onset, course, and duration of the diseases, the duration of the treatment, and the treatment effectiveness should also be recorded.

#### 3.1.2. Component 2: Multidisciplinary, Coordinated Team

Most experts agreed that the core professionals’ team should at least be composed of Maria’s GP, a geriatrician, a nurse, and a social worker. Other specialists such as endocrinologists, pulmonologists, cardiologists, clinical pharmacologists/pharmacists, psychiatrists, and psychologists could also integrate the team. The geriatrician was the preferred figure to lead the team, followed by the nurse and the GP. Clinical sessions and meetings, and a common electronic chart were the preferred communication tools for multidisciplinary team professionals.

#### 3.1.3. Component 3: Professional Appointed as Coordinator of the Individualized Care Plan and Contact Person

According to the experts, having a specific person as the primary contact to coordinate communications between Maria and the core team is crucial. All professionals of the care team must know who this coordinator is and who the final responsible care provider is. The coordinator needs to have good communication and organizational skills, has to be familiar with Maria’s medical and psychological situation, and must be knowledgeable on long-term care and community resources. This professional should be easy to reach, have frequent follow-up appointments, and should monitor whether provided care is in line with the wishes and needs of the patient. Most of the experts thought that the clinician and the contact person should be different professionals, where the former would be responsible for the somatic and physical problems, and the latter for the follow-up. Respondents reported that highly educated nurses with sufficient medical knowledge would be good candidates for the position of coordinator.

#### 3.1.4. Component 4: Individualized Care Plans

Experts identified the clinician as the most suitable person for the development of Maria’s care plan, always in close collaboration with other professionals. Maria’s care plan should address diabetes control, diagnosis and treatment of mental disorders, functional status improvement, and care arrangements for her and her husband. Plan revisions should occur in every visit, whether the desired goals and outcomes were obtained or not. Notwithstanding, one expert warned of the considerable administrative burden of devising such plans.

### 3.2. Decision Support

#### 3.2.1. Component 5: Implementation of Evidence-Based Practice

Specific clinical guidelines represent the best available knowledge for the conditions that Maria suffers from. Guidelines should adopt a patient-friendly perspective and consider her personal circumstances, conveying the importance of her participation in decision-making. Some examples applicable to Maria’s case included complex practice guidelines on multimorbidity, such as the Metabolic Vascular Syndrome of the Saxonian Chamber of Physicians [28] and the Slovenian type 2 diabetes guidelines [29], but also single-disease-oriented guidelines, such as the depression, multimedication, and back pain guidelines of the German Association of General Practice [30,31,32].

#### 3.2.2. Component 6: Training Members of the Multidisciplinary Team

Experts did not identify any specific training programs for the care team. Ideally, programs should include information on comprehensive care for multimorbidity and other important focus areas such as care prioritization, risk stratification, the patient’s needs and preferences, drug–drug interactions, the avoidance of polypharmacy, and the role, responsibilities, and limits of GPs as the gate-keepers of the health system. Additionally, other assets could be considered such as the understanding of roles and capacities of team professionals, the importance of the patient’s personal circumstances, values, and beliefs; teamwork skills; how to achieve agreed care plans; and even the understanding of human nature. In Maria’s case, experts suggested GPs, nurses, diabetologists, and clinical psychologists as good candidates for training programs.

#### 3.2.3. Component 7: Developing a Consultation System to Consult Professional Experts

For experts, the core team in the attention process should be the primary care GP and nurse duo, who could consider consulting other specialists under special circumstances that exceed their responsibilities or capacity to respond. Ideally, patient support groups, peer-supporters, and local patient associations could provide psychological support to patients and caregivers. Specialists should be consulted when primary care teams feel insufficient, when the criteria for referral have been met, or if therapeutic targets or the patient’s needs have not been reached. Existing guidelines and the needs of the patients should dictate the frequency of consultations. Several ways of providing access to specialists, offered by the experts, included phone calls or e-mails, face to face meetings, written consultations, or through patient associations.

### 3.3. Self-Management Support

#### 3.3.1. Component 8: Training of Care Providers to Tailor Self-Management Support Based on Patient Preferences and Competencies

Experts cited several existing training programs to help professional care providers improve their communication and self-management support skills. One example was a specific program for diabetes developed in the Netherlands to help care providers transform their disease-oriented vision into a more person-centered approach [33]. However, this program only focused on support for diabetic patients and did not consider multimorbidity as a whole.

Experts also emphasized the role that other care providers play in delivering tailored self-management support. Who provides said support depends on the nature of each condition or circumstance and the challenges it represents for self-management. For instance, a nurse could be the most appropriate in the case of diabetes management (e.g., self-monitoring and change of lifestyle habits), while homecare staff could provide advice for safety arrangements at home, and physiotherapists could offer support with physical activities for back problems.

#### 3.3.2. Component 9: Providing Options for Patients and Families to Improve Their Self-Management

For experts, the aspects of Maria’s health care plan that could be self-managed need to be agreed upon by both her and her care staff. When considering her options, experts emphasized the need to contemplate life-related factors (e.g., age, education level, health literacy, social circumstances, and network, ethnicity, lifestyle, preferences) and the barriers she may encounter for an adequate self-management, not only clinical diagnoses and medications.

Respondents identified a number of aspects that Maria could self-manage such as medication, diabetes monitoring, nutrition, pain relief, psychological or social support, making appointments with healthcare professionals, and caring for her husband. A thorough and empathic conversation should appraise her values, wishes, preferences, expectations, needs, possibilities, and ultimately result in a stepwise plan for achievable self-management activities. Experts agreed that Maria’s daughter could also attend a training program to improve her self-management skills. Such programs, like the Chronic Disease Self-Management Program, already exist in many countries [34]. Another example is the course ‘Beyond Good Intentions’, aimed at improving diabetes patients’ self-care and proactive coping skills in the Netherlands [35].

#### 3.3.3. Component 10: Shared Decision-Making

Maria, along with her daughter and husband, need to be invited to actively participate in decision-making by providing information on her current problems, thoughts, worries, and possible solutions. To do so, it would be helpful for her to prepare a list of questions regarding her health problems, what matters most to her, and what she expects from her visits. Maria ought to decide which family members partake in her care and the staff should interview them periodically and pay attention to their worries. Experts agreed that Maria’s care manager or the professional she trusts the most should be the one to inform and share decisions with her.

### 3.4. Information Systems and Technology

#### 3.4.1. Component 11: Electronic Patient Records and Computerized Clinical Charts

For experts, Maria’s clinical information should include a summarized overview of her conditions from each of the various medical teams, with regular updates on current treatments and possible side effects. Each team should update the information concerning the health issues they are responsible for accordingly. The record should provide a holistic and continuous view of the patients’ health as well as details of her treatment history and social support network.

#### 3.4.2. Component 12: Exchange of Patient Information between Care Providers and Sectors

Experts suggested providing patients, the care team, and an appropriate family member with access to health records, potentially increasing patient and caregiver support. Maria should be capable of restricting the access to her records to any appointed family member. Access control tools like passwords and PIN numbers were proposed, as well as encryption systems for stored information. As an example, the HIPAA Privacy Rule protects information from common security gaps that could lead to cyber-attacks or data loss. Using a Security Risk Assessment Tool would also be helpful, but any given security system should guarantee confidentiality (i.e., the privileged communication between two parties in a professional relationship) and privacy (i.e., the right of the individual patient to make decisions on how personal information is shared) [36].

#### 3.4.3. Component 13: Uniform Coding of Patients’ Health Problems

Experts highlighted the importance of using uniform coding systems to facilitate collaboration among professionals, and the clustering of patients based on clinical and organizational complexity. This strategy would maximize the efficacy and cost-effectiveness of interventions and ensure greater patient safety. Implementing risk stratification tools to tailor practices to the specific needs of patients could also prove helpful. Several coding and/or classification systems could be used here, such as the International Classification of Primary Care, the Adjusted Morbidity Groups, or the International Classification of Functioning, Disability and Health [37,38,39].

#### 3.4.4. Component 14: Patient-Operated Technology Allowing Patients to Send Information to Care Providers

Experts considered that Maria would be able to use technologies if adequately motivated. The core team should actively motivate her and her caregivers through self-management support, shared decision-making, and education/information, taking into account her social and economic situation. These technologies would require periodic re-evaluation programs to ensure patients keep making adequate use of them. Regarding diabetes monitoring, experts suggested using several wearable devices such as patches, pre-loaded medication packs, or equipment to self-monitor blood glucose and blood pressure levels. Mobile applications with glucose diaries, patient platforms with video and/or audio tools, as well as sleep-monitoring technologies were some of the other options they offered.

### 3.5. Social and Community Resources

#### 3.5.1. Component 15: Supporting Access to Community and Social Resources

To facilitate Maria’s access to community and social resources, experts recommended: Better housing (e.g., availability of an elevator), nutritional support, connecting Maria with relevant activities in her community, and reinforcing her social contacts. Primary care professionals should advise Maria to get in contact with workers from her municipality such as the social worker at the city/town council. The initial participation of the case manager in this area is crucial. He/she should coordinate all efforts with social workers to detect Maria’s needs and provide her with information on the services available to her. The most notable community and social resources identified as suitable for this case were home support for activities of daily living (e.g., housework, shopping, personal hygiene), telecare, dependency assessment, financial support, and day-care centers for her husband.

#### 3.5.2. Component 16: Social Network Involvement

Although Maria’s daughter should be the first person to get more involved in her mother’s care, she would need a better understanding of her family’s current situation and social relationships to do so effectively. Neighbors could also be helpful in specific situations, especially in ‘raising the alarm’ should they notice anything wrong. Local organizations such as her parish or local charities could also provide support. The level of involvement in Maria’s case expected from each person should be set according to his/her desires, possibilities and capacities, after reaching an agreement with the GP, social worker and/or care coordinator. The case manager and the social worker were proposed as the professionals responsible for involving Maria’s social network in her care.

## 4. Discussion

Patients with multimorbidity have complex needs and their care involves a wide variety of healthcare providers and resources. However, research on interventions for multimorbidity remains scarce [3], and there are very few specific strategies to improve the management of patients suffering from this increasingly prevalent condition [39,40]. The JA-CHRODIS IMCM proposes a multidimensional approach for the care of patients with multimorbidity based on the consensus of European experts. The case designed for study provides a suitable framework in which to describe in detail the potential implementation of the IMCM. This work aims to support the usage of the model in clinical practice by identifying relevant barriers and recommendations for the implementation of each component.

Supporting policy makers in the management of people with chronic conditions and their emerging needs is a challenge that various care models, such as the Guided Care model and Wagner’s Chronic Care Model [41,42], had already attempted to address. The IMCM was built upon the foundations set by those models and is based on the same underlying principles, structured into five dimensions (i.e., delivery of care, decision support, self-management support, community resources, and information systems). Despite that, the IMCM is considered a living model, distinctive for its adaptability and subject to the addition of new elements by the CHRODIS group as the opportunities to do so arise. For instance, experts are currently incorporating a new dimension with the objective of improving employment access for people with chronic diseases and supporting employers to promote healthy activities for the prevention of chronic diseases in the workplace [43]. In this sense, good practices regarding employment management for people with chronic diseases have been developed, creating pathways to optimize employment prospects and working conditions. Some of these practices consist of integrative support services that offer coherent pathways for people with chronic conditions to foster their staying-in, integration, or reintegration in the labor market; other practices are based on rehabilitation programs, including work-life related psycho-social support, for which labor market participation represents a key goal [43]. Future versions of the model integrating this new dimension on employment and chronic diseases should be reevaluated regarding their potential applicability.

Numerous studies suggest that multimorbidity interventions need to be integrated into existing healthcare systems to support their implementation [42,44,45,46]. Our work evaluates the applicability and transferability of the IMCM and offers insight from experts from various countries to identify key factors for its promotion and integration in different healthcare systems and scenarios. Notwithstanding, local adaptations will likely be necessary even for interventions that are effective in other specific contexts. For example, the Cochrane review showed that interventions targeting comorbid depression, although effective, require training and support for primary care professionals, which may not be available in every setting [9].

The most recent Cochrane Review, focusing on patient-level approaches to multimorbidity management [9], suggested that health outcomes improve when interventions are targeted to population groups with specific risk factors (e.g., depression, specific functional difficulties). Certain studies of the review suggested that patient-level interventions had limited impact if performed in isolation, concluding that multimorbidity care models with ’whole-system organization’ approaches would be more effective. The opinions gathered for our research reassert the importance of this holistic approach and our analysis found many experts, despite their different profiles, concurring in the use of the same conceptual elements such as “support”, “information”, “contact”, or “team”.

Fragmentation of care due to the involvement of multiple care professionals without effective communication represents a real problem for patients with multimorbidity. In this case study, Maria requires integrated interventions from several professionals, where communication among team professionals and the existence of a known contact acting as care coordinator are crucial to avoid care fragmentation. The implementation of the model, as showed in the case study, requires the use of a wide array of rating scales and tools to assess patient needs in a comprehensive and regular way. These instruments could be helpful not only for comprehensive assessments, but also for the coordination between health and social services, which is crucial to perform patient-centered integrated care.

Clinical guidelines that offer decision-making support adapted to multimorbidity should focus on patients’ wishes, beliefs, and needs, and include chapters on concordant and discordant diseases. Healthcare professionals, however, often perceive that they lack specific trainings to work as a team or to address the needs of patients with multimorbidity and their caregivers [46]. Developing consultation systems to contact external experts would be a useful asset to support decision-making, however, these systems should be timely and flexible to facilitate their implementation and allow for the appropriate exchange of information.

The distinctive features of the different health systems from each country or region (e.g., single or multiple care providers, type of financing mechanisms, decentralization of management of care delivery, level of integration development, or coordination procedures) could limit the development or implementation of key aspects necessary for the model to work. Therefore, analyzing from different perspectives which sections/components of the model can be implemented, and the adjustments that would be necessary to do so in each context, will be essential for an optimal implementation. In this sense, JA-CHRODIS-PLUS is currently performing a pilot implementation of the model in five European care settings [47], and one of the main objectives, besides the overall assessment of its applicability in clinical practice, is to provide country-specific integrated care model versions with local adaptations taking into consideration local features.

Currently, several actions throughout Europe identify two crucial features when attending complex cases like Maria’s: A multidisciplinary team consisting of primary and specialized healthcare professionals, social workers, and engaged family members; and the necessity of a designated case manager. The clustering of patients based on clinical and organizational complexity is also essential to maximize the efficacy and cost-effectiveness of interventions and ensure greater patient safety. Implementing risk stratification tools may also allow tailoring practices to the individual contexts and needs of patients.

One of the main limitations of the study lies in the limited number of expert opinions used to assess the applicability of the model. Moreover, an unequal number of experts analyzed each component, and, in some cases, results were based on the responses from only two experts. Their different backgrounds and/or variable degree of expertise could have potentially biased the information obtained for each component. This study represents a preliminary assessment of the model´s applicability in clinical practice, and future studies are encouraged to assess the model based on a greater number of opinions and to evaluate the potential applicability in different healthcare settings and countries, in line with the pilot implementation that is being conducted in the context of JA-CHRODIS-PLUS.

The results of this qualitative study showed, through Maria’s case, that the IMCM can provide a flexible framework to be applied in different contexts for the delivery of patient-centered care in chronic patients.

## 5. Conclusions

Our results suggest that the JA-CHRODIS IMCM is potentially applicable in a complex multimorbidity case of a person with diabetes, mental health issues, and several psychosocial problems, providing a favorable framework to deliver person-centered care for patients with multimorbidity. Experts concurred that elements such as support, teamwork, and information should be the cornerstones of the attention process for chronic patients. Pilot studies with real cross-national applications of the JA-CHRODIS IMCM, as the ones developed in JA-CHRODIS-PLUS, are called for.

## Figures and Tables

**Figure 1 ijerph-16-05151-f001:**
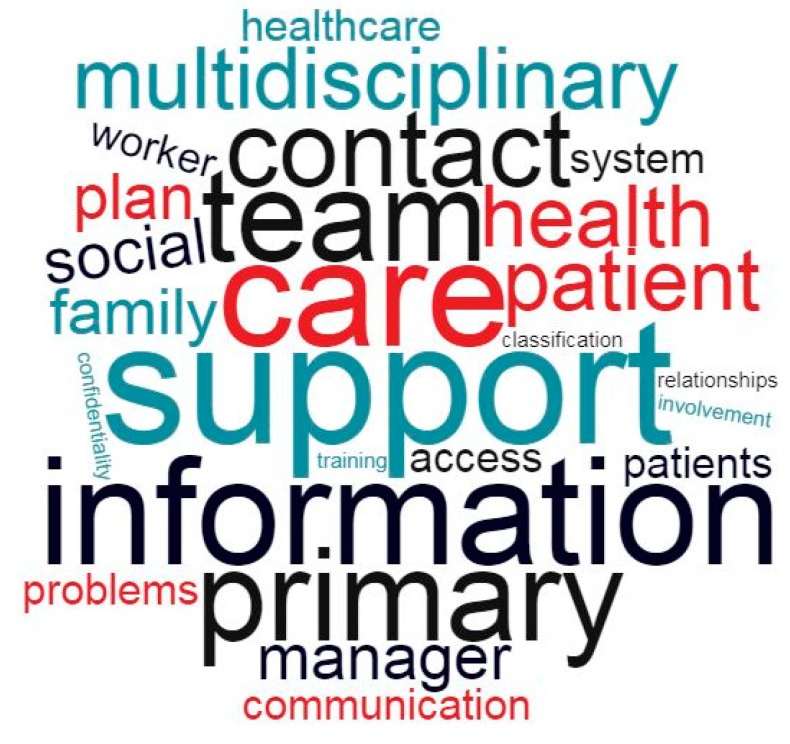
Word cloud chart merging the five sections of the Integrated Multimorbidity Care Model.

**Table 1 ijerph-16-05151-t001:** Distribution of the Integrated Multimorbidity Care Model components evaluated by participating experts from different European countries.

	Components of the Integrated Multimorbidity Care Model ^1^															
Country (Number of Experts)	1	2	3	4	5	6	7	8	9	10	11	12	13	14	15	16
Croatia (1)			x	x												
Italy (1)	x	x											x	x		
Germany (1)					x	x	x				x	x				
Lithuania (1)	x	x														
Netherlands (2)			x	x				x	x	x					x	x
Slovenia (1)					x	x	x									
Spain (1)															x	x
United Kingdom (3)								x	x	x	x	x	x	x		

^1^ 1: Regular assessment of patients; 2: Multidisciplinary team; 3: Case manager; 4: Individualized care plans; 5: Evidence based practice; 6: Training; 7: Consultation system; 8: Training of care providers to tailor self-management support; 9: Providing options for patients and families; 10: Shared decision making; 11: Electronic patient records; 12: Exchange of patient information; 13: Uniform coding; 14: Patient-operated technology; 15: Community and social resources; 16: Involvement of social network.

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
