# Peer review of "Application of the JA-CHRODIS Integrated Multimorbidity Care Model (IMCM) to a Case Study of Diabetes and Mental Health"

_ijerph, 2019, doi:10.3390/ijerph16245151_

Round 1

Reviewer 1 Report

My comments are provided in the attached PDF file.

Reviewer 2 Report

Dear Authors,

Your study on Application of JA-CHRODIS Integrated Multimorbidity Care Model is very interesting and good paper. The paper is generally well structured and I only have some minor comments that authors should address before publication in the Journal.

Discussion and conclusion:

In future prospective of model,  I suggest to deepen the new dimension of IMCM (improving employment access for people with chronic diseases) and to propose new practices.  For example, in the care pathways there could be 2 possible ways of introducing the issue of employment: the first is along all the care pathways, the second is within one of the phases of the rehabilitation program. The first way, from primary care to specialized interventions there should be integration so as to have a coherent pathway for a person with a chronic condition able to foster staying in, integration or reintegration in the labor market. To do this it is of most importance to identify simple ways to monitor work ability in the working population on a regular basis.  In the Rehabilitation programs, Labour market participation represents a key goal of rehabilitation for individuals with chronic conditions. A standard instruments in the rehabilitation part of the multimorbidity chronic care model could be useful to standardize pathways and make them less fragmented for people with NCDs.
